# Metabolic profiling of cancer cells reveals genome-wide crosstalk between transcriptional regulators and metabolism

Karin Ortmayr [1], Sébastien Dubuis[1] & Mattia Zampieri [1]

Transcriptional reprogramming of cellular metabolism is a hallmark of cancer. However, systematic approaches to study the role of transcriptional regulators (TRs) in mediating cancer metabolic rewiring are missing. Here, we chart a genome-scale map of TR-metabolite associations in human cells using a combined computational-experimental framework for large-scale metabolic profiling of adherent cell lines. By integrating intracellular metabolic profiles of 54 cancer cell lines with transcriptomic and proteomic data, we unraveled a large space of associations between TRs and metabolic pathways. We found a global regulatory signature coordinating glucose- and one-carbon metabolism, suggesting that regulation of carbon metabolism in cancer may be more diverse and flexible than previously appreciated. Here, we demonstrate how this TR-metabolite map can serve as a resource to predict TRs potentially responsible for metabolic transformation in patient-derived tumor samples, opening new opportunities in understanding disease etiology, selecting therapeutic treatments and in designing modulators of cancer-related TRs.

---

[1] Institute of Molecular Systems Biology, ETH Zurich, Otto-Stern-Weg 3, CH-8093 Zurich, Switzerland. Correspondence and requests for materials should be addressed to M.Z. (email: zampieri@imsb.biol.ethz.ch)

Transcriptional regulators (TRs) are at the interface between the cell's ability to sense and respond to external stimuli or changes in internal cell-state[1]. In cancer as well as other human diseases[2], alterations in the activity of TRs, such as transcription factors, chromatin modifiers or transcription factor co-regulators, can remodel the cellular signaling landscape and trigger metabolic reprogramming[3] to meet the requirements for fast cell proliferation and cell transformation[4,5]. However, evidence linking alterations of cancer metabolism to TR dysfunction is often based on molecular profiling technologies, like transcriptomics and chromatin modification profiling[6] or the identification of TR-binding sites upstream of metabolic enzymes[3], that don't report on the functional consequences of detected interactions. Mass spectrometry-based metabolomics approaches are powerful tools for the direct profiling of cell metabolism and to uncover mechanisms of transcriptional (in)activation of metabolic pathways[7,8]. Nevertheless, because of limitations imposed by commonly used workflows, such as coverage, scalability, and comparability between molecular profiles of largely diverse cell types, simultaneously quantifying the activity of TRs and metabolic pathways at genome- and large-scale remains a major challenge. Here, we develop a unique experimental workflow for the parallel profiling of the relative abundance of more than 2000 putatively annotated metabolites in morphologically diverse adherent mammalian cells. This approach overcomes several of the major limitations in generating large-scale comparative metabolic profiles across cell lines from different tissue types or in different conditions, and was applied here to profile 54 adherent cell lines from the NCI-60 panel[9].

To understand the interplay between transcriptional regulation and emerging metabolic phenotypes, we generated a genome-scale map of TR-metabolite associations. To this end, we implemented a robust and scalable computational framework that integrates metabolomics profiles with previously published transcriptomics[10] and proteomics[11] datasets to resolve the flow of signaling information across multiple regulatory layers in the cell. This computational framework enables (i) systematically exploring the regulation of metabolic pathways, (ii) reverse-engineering TR activity from in vivo metabolome profiles and (iii) predicting post-translational regulatory interactions between metabolites and TRs. Beyond contributing to the understanding of genome-wide associations between changes in TR activities and rewiring of metabolism in cancer, we demonstrate how genome-scale TR-metabolite associations can introduce a new paradigm in the analysis of patient-derived metabolic profiles and the development of alternative therapeutic strategies to counteract upstream reprogramming of cellular metabolism.

## Results

**Large-scale metabolic profiling of cancer cells**. Tumor cells, in spite of similar genetic background or tissue of origin, can exhibit profoundly diverse transcriptional and metabolic phenotypes[12,13]. Exploiting the naturally occurring variability across a large set of diverse cancer cell lines could reveal the interplay between aberrant tumor metabolism and gene expression. However, in spite of significant advancements in the rapid generation of high-resolution spectral profiles of cellular samples[14,15], the accurate comparative profiling of intracellular metabolites across heterogeneous cell line panels is still a major challenge. Two are the major bottlenecks for in vitro large-scale cell metabolic profiling: (i) limited throughput of classical techniques[16] featuring large cultivation formats, time-consuming and laborious extraction/measuring protocols, and (ii) lack of accurate and rapid normalization procedures to make metabolic profiles of morphologically diverse cell types comparable. This last step is often

implemented by the additional quantification of total protein abundance and is based on the assumption that protein content scales with cell volumes, even across largely diverse cell types or conditions[17] (Supplementary Figs 1, 2). To overcome these limitations, we present an innovative and robust workflow enabling large-scale metabolic profiling in adherent mammalian cells alongside with a scalable computational framework to normalize and compare molecular signatures across cell types with large differences in morphology and size (Supplementary Fig. 1–2). In contrast to classical metabolomics techniques[16], we use a 96-well plate cultivation format, rapid in situ metabolite extraction, automated time-lapse microscopy and flow-injection time-of-flight mass spectrometry[14] (FIA-TOFMS) for high-throughput profiling of cell extract samples (Fig. 1a).

Specifically, we optimized each step, from cultivation and extraction to MS analysis, to be compatible with parallel 96-well processing. Different cell lines are seeded in triplicates at low cell density in 96-well microtiter plates, and are grown to confluence within 5 days (37 °C, 5% $CO_2$). Growth is continuously monitored by automated acquisition of bright-field microscopy images, and replicate 96-well plates are sampled for metabolome analysis every 24 h. To facilitate sampling, increase the throughput and reduce the risk of sample processing artifacts, we collect metabolomics samples directly in the 96-well cultivation plate without prior cell detachment (Supplementary Fig. 1). Finally, cell extracts are analyzed by FIA-TOFMS[14], allowing rapid full-spectral acquisition within less than one minute per sample.

Normalization of large-scale non-targeted metabolomics profiles is a fundamental aspect of data analysis, which becomes particularly challenging when comparing mammalian cell lines with large differences in cell size. Our approach consists of two main steps. First, we quantified relative metabolite abundance per cell using a multiple linear regression scheme. To this end, we quantified cell numbers in each sample directly from automated analysis of bright-field microscopy images (see Supplementary Note and Supplementary Fig. 1), and for each cell line, related extracted cell number to ion intensity (Fig. 1b). By combining MS profiles throughout cell growth (Supplementary Fig. 1), we decoupled cell line-specific metabolic signatures from differences in extracted cell numbers. This procedure enables selecting only annotated ions exhibiting a linear dependency between measured intensities and extracted cell number (Fig. 1b), that are hence amenable to accurate relative quantification. Moreover, by integrating MS readouts at multiple cell densities and time points, the resulting estimates of relative metabolite abundances are invariant to cultivation time and cell densities, enabling the direct comparison between different cell lines and conditions[18]. In the second step, a subset of metabolites identified to report on cell volume, mostly intermediates of fatty acid metabolism, were used to correct for differences in total volume of sampled cultures (see Methods section and Supplementary Fig. 2).

**Linking gene expression profiles to metabolic diversity**. Here, we used our framework to profile the intracellular metabolomes of 54 adherent cell lines from eight different tissue types in the NCI-60 cancer cell line panel. For 2181 putatively annotated ions exhibiting a significant linear dependency between extracted cell number and ion intensities (linear regression $p$-value ≤ 3.4e−7, Bonferroni-adjusted threshold), we report Z-score normalized relative abundances fitted over approximately 15 individual measurements per cell line (Fig. 1b and Supplementary Data 1). This new data set illustrates the technological advances of our methodology over previous metabolomics approaches and similar comparative resources (Supplementary Fig. 3), in that it enables

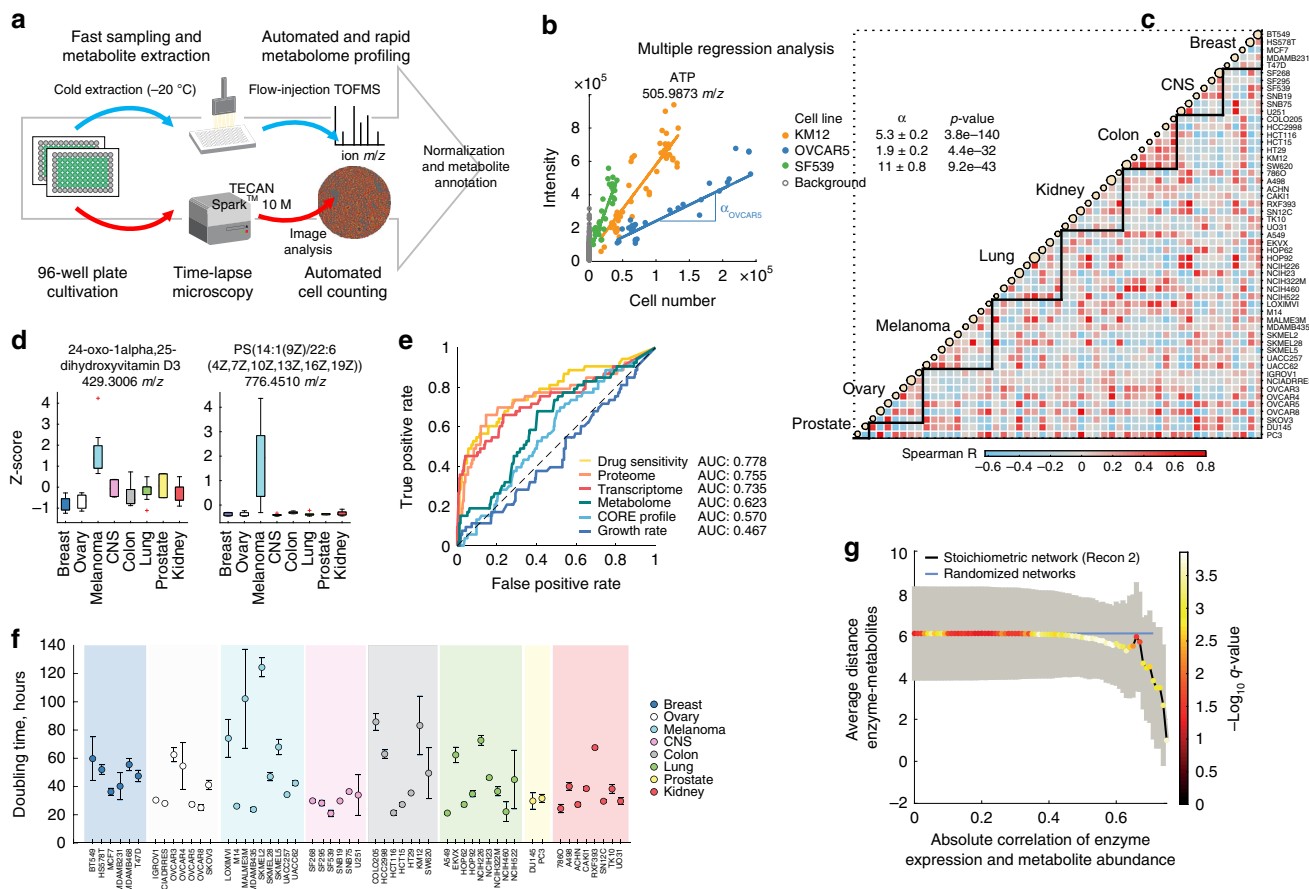

**Fig. 1** Comparative metabolome profiling in 54 adherent cancer cell lines. **a** Schematic overview of the combined workflow for high-throughput metabolome profiling in adherent cell lines. Multiple cell lines are cultivated in parallel to collect cell extracts for MS-based metabolome profiling (Supplementary Fig. 1). A new software tool for the segmentation of bright-field microscopy images (Supplementary Note) allows automated cell number quantification. **b** Raw MS data for adenosine triphosphate (ATP) in three different cell lines before cell volume correction. Slopes (α) and standard errors from the linear fitting of ion intensity and cell number are reported, together with the p-value significance. Background intensities were obtained from measurements of cell-free extraction solvent. **c** Pairwise similarity (Spearman correlation) of metabolome profiles among 53 cell lines from the NCI-60 panel. Circle size on the diagonale is proportional to cell line doubling times (**f**). **d** Examples for metabolites exhibiting a significantly (ANOVA q-value ≤ 0.05, corrected for multiple tests) tissue-dependent abundance (Supplementary Fig. 4, Supplementary Data 1). Boxplot of Z-score normalized metabolite levels are grouped by the cell line tissue of origin. Box edges correspond to 25th and 75th percentiles, whiskers include extreme data points, and outliers are shown as red plus signs. **e** Signatures of tissue type in intracellular metabolome profiles compared to transcriptome[10], proteome[11], drug sensitivity[31], growth rate and the uptake- and secretion rates of 140 metabolites (CORE profiles[13]). Receiver operating characteristic curve analysis quantifies the likelihood of cell lines from the same tissue type to feature similar molecular profiles. **f** Cell line doubling times (mean ± standard deviation), grouped by tissue. Doubling times were determined from continuous monitoring of cell confluence in 96-well plates. **g** Correlations between metabolite levels and gene expression in relation to their distance in the stoichiometric network of human metabolism[21]. The black line represents the average pairwise enzyme-metabolite distance (y-axis) at different levels of correlation between metabolite abundance and enzyme gene expression (x-axis). The blue line represents the average enzyme-metabolite distance in 10.000 randomized networks. Dot color reflects significance (q-value from permutation test) of enzyme-metabolite proximity in the stoichiometric network as compared to the randomized networks

higher throughput, sensitivity, coverage and systematic removal of biases associated to differences in cell size.

Pairwise similarity analysis of cell line metabolome profiles (Fig. 1c) reveals widespread heterogeneity in the metabolome of cell lines even from the same tissue type. We found only 70 metabolites whose intracellular levels exhibited a significant dependency (ANOVA, q-value ≤ 0.05) on cell-line tissue (Fig. 1d, Supplementary Fig. 4, Supplementary Data 1). In few cases, these patterns highlighted expected tissue-specific functions, such as for elevated levels of a derivative of vitamin D3 in melanoma cells (Fig. 1d), while other metabolites were ubiquitously present among cell lines but in different levels depending on the tissue of origin. While previously published transcriptome[10] and proteome[11] profiles for 53 of the herein profiled cell lines revealed a stronger molecular signature of the tissue of origin

(Fig. 1e), heterogeneity in metabolome profiles across cell lines from the same tissue type is consistent with large differences in doubling times (Fig. 1f, Supplementary Fig. 4, Supplementary Data 1), previous data on exchange rates of nutrients and metabolic byproducts[13,19] (Fig. 1e) and metabolic differences across mouse tissues[20] (Supplementary Fig. 3).

How can such large phenotypic and metabolic diversity emerge under the same environmental condition? We hypothesized that differential regulation of gene expression could subserve metabolic heterogeneity. To test our hypothesis, we correlated mRNA levels of metabolic enzymes with metabolite abundances, and related enzyme-metabolite correlation to their proximity in the metabolic network. We used a genome-scale stoichiometric model of human metabolism[21] to derive the distance between each enzyme-metabolite pair as the minimum number of

reactions separating the two in the network. We found that enzyme gene expression tends to more strongly correlate with levels of proximal metabolites (Fig. 1g), reflecting a direct dependency between enzyme- and related metabolite levels[22,23]. By expanding the search for correlation between gene expression- and metabolite levels beyond enzyme-encoding genes, we observed that the two principal components of metabolic variance explaining 38% of total variance (Supplementary Fig. 4) most strongly correlated (Spearman $|R| > 0.37$) with transcripts in signal transduction pathways regulating cell proliferation, adaptation, cell adhesion and migration (e.g., HIF-1, PI3K-Akt, AMPK, Supplementary Fig. 4). Altogether, these results suggest that phenotypic heterogeneity observed in vitro can emerge as a result of different transcriptional regulatory programs, and that metabolite abundances can be used as intermediate functional readouts linking gene expression profiles to different strategies in allocating metabolic resources for energy generation and growth.

**Systematic inference of TR-metabolite associations.** To study the flow of signaling information between transcriptome and metabolome, we sought to quantify the functional interplay between metabolic phenotypes and different transcriptional programs mediated by the activity of transcriptional regulators (TRs). Here, we use the term TR formally for any regulator capable of modulating gene expression, including transcription factors, chromatin remodelers, and co-regulators, i.e., adopting the inclusion criteria from a curated repository of gene regulatory network links[24] (TRRUST database). TRs can directly regulate metabolic fluxes by modulating enzyme abundance, i.e. changing maximum flux capacity, or by indirectly affecting substrate availability of proximal metabolic reactions, which can in turn result in local changes of fluxes[25,26]. The activated form of a transcriptional regulator, rather than its expression/protein level, regulates gene promoters, and a TR's activity is imprinted in the expression levels of its target genes. Hence, because TR activity is governed by complex post-transcriptional and post-translational mechanisms, monitoring TR gene expression or protein levels is an inadequate proxy of their activity[27,28] (Supplementary Fig. 5). Because, methodologies to directly measure promoter activity, such as GFP or luciferase constructs, are limited in scalability, we used network component analysis (NCA)[29] to derive a relative estimate of TR activity for each cell line directly from the combined expression levels of TR-gene targets. By quantifying TR activities from transcriptomic profiles and a network model of transcriptional regulation, NCA enables simultaneously comparing the activity of TRs across different cell lines on a genome-scale level.

Hence, by integrating previously published transcript abundance data[10] for 53 cell lines with a genome-scale network of literature-curated TR-target gene interactions (TRRUST database[24]), we derived a relative estimate of TR activity for 728 transcriptional regulators (Fig. 2a, Supplementary Fig. 5). To account for potentially confounding effects of different growth rates and the incompleteness of the TR-regulatory network, we introduced an artificial TR that, by virtually targeting all genes, simulates pleiotropic effects of growth on gene expression, and used a bootstrapping approach to sample different combinations of TR regulatory interactions. Within an unknown scaling factor, TR activities are hence robustly derived as the median across more than 400 estimates per TR (Supplementary Fig. 5, Supplementary Data 2).

To generate an empirical network of associations between TRs and metabolites, we systematically correlated the activity of 728 TRs with relative levels of individual metabolites across cell lines (Fig. 2a, Supplementary Data 3, Supplementary Fig. 5). First, we investigated whether metabolites correlating with TR activity locate in the proximity of TR-enzyme targets. To this end, we estimated the distance of each TR to metabolites by taking the minimum distance between known[24] TR enzyme-targets and metabolites in the metabolic network. We found that metabolites exhibiting the strongest correlations with TR activity (|Spearman correlation| > 0.45) are in the significant (q-value < = 0.05, permutation test) vicinity of TR-enzyme targets (Supplementary Fig. 5). Such local dependencies support our hypothesis that metabolite levels can be used as an intermediate readout to study the functional interplay between regulation of gene expression and cell metabolism.

To test the extent to which our conclusions hold true beyond the specific TR-regulatory network and cell lines tested here, we expanded the TR-gene target network to include any enzyme whose transcript levels correlate with TR activity, i.e. enzymes directly or indirectly regulated by a TR. To that end, we applied NCA to resolve TR activity across an independent large compendium of transcriptome data, monitoring gene expression in a panel of 1037 human cancer cell lines[30] (Cancer Cell Line Encyclopedia), and identified enzyme levels that correlate with individual TR activity profiles (|Spearman correlation| > 0.5, Supplementary Data 6). We call the resulting association network an augmented TR-gene network. By repeating the analysis of TR-metabolite distance using the augmented TR-gene network, we found an even stronger vicinity between TRs and correlating metabolites (Fig. 2b). Hence, the analysis of this independent panel of cancer cell line expression data not only reinforces our previous results, but it also suggests that the TR-metabolite association network derived from the NCI-60 tumor cell lines can be generalized to a largely diverse panel of cell lines. Our results demonstrate that while models of TR-target genes are far from complete, even relatively few known gene targets can be sufficient to estimate TR activities. Remarkably, while in human cells most of the known TR binding sites map to genes in signaling- and disease-related pathways, our TR-metabolite network unraveled a complementary large space of associations involving intermediates in central metabolic pathways (Fig. 2c).

**Mapping TR activity to metabolic phenotypes.** Here, we asked whether coordinated changes in TR activity and metabolite abundances indirectly inform on changes in proximal metabolic fluxes[12] (see Supplementary Fig. 6 and Supplementary Discussion). As a proof of concept, we measured rates of glucose uptake and lactate secretion in each cell line as a proxy for glycolytic flux. As previously observed[13,22], glucose consumption strongly correlated with lactate secretion, and on average approximately 70% of the incoming glucose carbon is secreted into lactate (Fig. 2d and Supplementary Fig. 3). Next, we related intracellular metabolite abundances to measured fluxes by estimating the average correlation with glucose and lactate exchange rates (Fig. 2e). Indeed, metabolites that strongly correlate with glycolytic flux (Spearman $R > 0.5$) were enriched for intermediates of proximal glycolytic pathways, oxidative phosphorylation and HIF-1 signaling pathway (Fig. 2f), including metabolites such as glyceraldehyde 3-phosphate, acetyl-CoA and ATP (Fig. 2e). Surprisingly, metabolites that anti-correlated with glucose uptake, i.e. that are increased with low glycolytic flux, were significantly (hypergeometric test, q-value ≤ 0.001) enriched for one-carbon metabolism (Fig. 2f), hinting at a direct functional dependency between one-carbon metabolism and glycolytic flux. To test this prediction, we correlated glucose consumption and lactate secretion with the susceptibility to 430 drugs with known mechanisms of action[31]. Consistent with the hypothesized functional dependency between glucose and one-carbon metabolism,

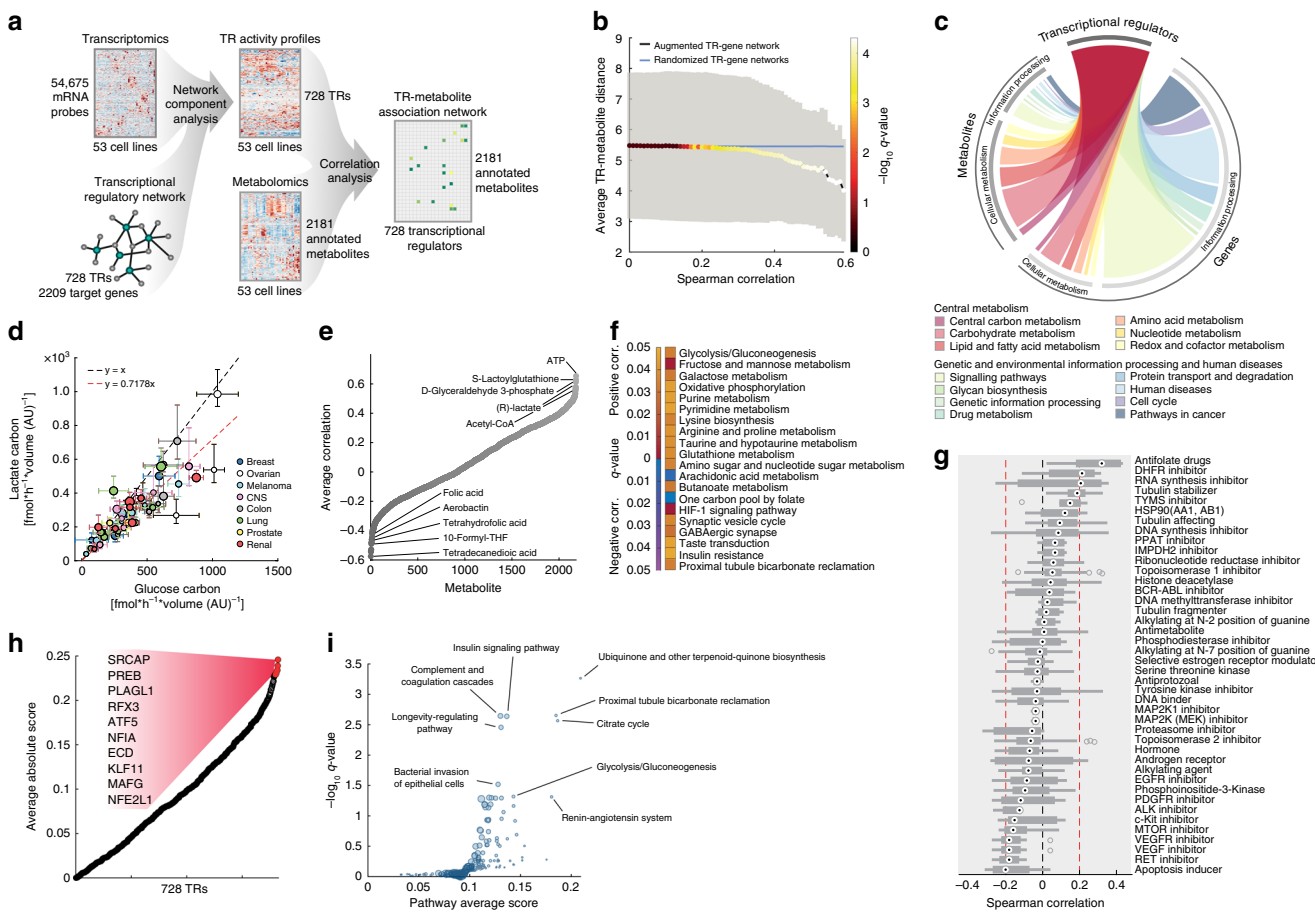

**Fig. 2** Inferring a TR-metabolite association network by integrating transcriptome and metabolome data. **a** Schematic overview of the computational framework for finding TR-metabolite associations. Activity of 728 human TRs was calculated using Network Component Analysis[29] on expression profiles of the NCI-60 cell lines[10] and the TRRUST transcriptional regulatory network[24]. Correlation analysis was used to find associations between TRs and metabolites. **b** Correlations between metabolite levels and TR activity profiles in relation to the distance between TRs and enzyme targets in the metabolic network[21]. The black line represents average TR-metabolite distance (y-axis) at different levels of absolute TR-metabolite correlation (x-axis). The blue line represents average TR-metabolite distance in 10.000 randomized networks. Dot color indicates significance (q-value, corrected for multiple tests) of TR-metabolite proximity compared to the randomized networks. **c** Distribution of TR-metabolite associations (0.1% FDR, left section) and TR-gene regulatory interactions (right section) across KEGG pathways. Edge-size connecting the TR hub and metabolic pathways reflects the number of links found in either network. **d** Glucose uptake vs lactate secretion. Mean ± standard deviation of rates estimated over three biological replicates and five time points. **e**, **f** Metabolites whose abundance correlated with glycolytic flux (i.e., glucose uptake and lactate secretion, **e**) were tested for significant (q-value, corrected for multiple tests) enrichment in KEGG metabolic pathways (**f**), separately for positively and negatively correlated metabolites (yellow to red and purple to blue color ranges, respectively). **g** Boxplot of the correlation between glucose uptake/lactate secretion and sensitivity (i.e., GI₅₀, drug concentration causing 50% growth rate reduction) to 430 drugs[31] grouped by mechanism of action. Box edges reflect 25th and 75th percentiles, center dots indicate medians, whiskers include extreme data points, and outliers are gray circles. **h** Association between TR activity and metabolites reporting on glycolytic flux. TR names are shown for the 10 TRs with the highest score. **i** For each KEGG pathway we calculated the average association score (**h**) among TRs with known enzyme targets in the pathway (x-axis). Significance (y-axis) is estimated using a permutation test and corrected for multiple tests (q-value). Only pathways with a q-value ≤ 0.01 are labeled

we found that among all drug classes, the strongest correlation with glycolytic flux was observed with cell line sensitivities to antifolate drugs and inhibitors of dihydrofolate reductase (DHFR) —i.e., cell lines with lower glucose uptake are more sensitive to inhibitors of folate biosynthesis (Fig. 2g).

Next, we sought to use our TR-metabolite network to find TRs associated with glycolytic flux, that potentially coordinate glucose and one-carbon metabolism. To this end, we searched for TRs that correlate with metabolites reporting on glycolytic flux by calculating the sum of the dot product between TR-metabolite- and metabolite-glycolytic flux correlation vectors, normalized by the absolute sum of TR-metabolite correlation coefficients (Fig. 2h). Notably, TRs with the highest scores were enriched (permutation test, q-value ≤ 0.05) for enzyme targets in

ubiquinone biosynthesis, insulin signaling, TCA cycle and glycolysis/gluconeogenesis (Fig. 2i). Several of the top 10 predicted TRs (Fig. 2h) are associated to the regulation of key steps in glycolysis, such as the regulatory factor X3 (RFX3) regulating the glucokinase gene[32], the nuclear factor erythroid 2-related factor 1 (NFE2L1) and its interacting partner MAFG, involved in the regulation of oxidative stress response and diverse glycolytic genes[33]. Interestingly, we also found TRs important for the survival and proliferation of cancer cells under nutrient limitation or stress, such as the activating transcription factor 5 (ATF5)[34] and the SNF2-related CPB activator protein (SRCAP), a direct regulator of phosphoenolpyruvate carboxykinase 2 (PCK2)[35], a gluconeogenic enzyme essential to maintain cell proliferation under limited glucose conditions in cancer cells[36].

Consistent with our previous observation of a dependency between glycolytic flux and intermediates in one-carbon metabolism, SRCAP activity levels across the 1037 CCLE cancer cell lines strongly correlate ($R \geq 0.5$) with the expression of three genes (Supplementary Data 2): MTHFD2, a mitochondrial enzymes in folate metabolism, asparagine synthetase (ASNS) and ATF4, an upstream regulator of serine biosynthesis and mitochondrial folate enzyme transcription, involved in the response to amino acids starvation (Supplementary Fig. 6).

It is tempting to speculate that cancer cell line diversity in glucose uptake and lactate secretion observed in vitro potentially reflects regulatory programs acquired during an earlier adaptation to in vivo nutrient availability and stresses. Altogether, these findings independently support the functional relevance of our TR-metabolite association network in resolving the interplay between transcriptional regulation and metabolic phenotype. Notably, the inferred associations between metabolic intermediates and TRs are complementary to the information derived from TR-gene regulatory networks. While the latter report on the regulatory capability of TRs to change enzyme abundance, coordinated changes between TR activity and metabolite levels can reveal the functional implications of these regulatory events.

**Predicting functional roles of TRs in metabolism.** Because correlation does not imply causation, we cannot resolve whether changes in metabolite levels are responsible for changes in TR activity or vice-versa. However, often only few metabolic intermediates of metabolic pathways, typically the end product[37], can allosterically regulate TRs. Hence, multiple proximal metabolic intermediates correlating with an individual TR's activity can reveal the functional impact of TRs on overall pathway activity. By analyzing TR-metabolite associations on pathway-level, we can hence predict the functional roles of individual TRs in potentially regulating distinct metabolic pathways. For 677 TRs, we discovered a significant enrichment (hypergeometric test, $q$-value $\leq 0.05$) of TR-associated metabolites in at least one KEGG pathway (Fig. 3a, Supplementary Fig. 6, Supplementary Data 4), including 145 cancer-related TRs[38] (as defined by the COSMIC cancer gene census). In this analysis, metabolic pathways with the highest number of associated TRs were arachidonic and fatty acid metabolism, followed by arginine and proline metabolism and the degradation of branched-chain amino acids (Supplementary Fig. 6). These predicted interactions potentially reflect a large space of yet unexplored regulatory interactions that can mediate the adaptation to varied micro-environmental conditions and fast proliferation under diverse nutrient availability (see Supplementary Discussion).

Even in cases where the roles of TR-target genes have been extensively characterized, the herein-proposed TR-metabolite associations can help refining the condition-specific functional role of TRs in metabolism. For example, hypoxia-inducible factor 1 alpha (HIF-1A)[39] is reported to act on regulatory elements upstream of nearly 50 enzymes in several central metabolic pathways (Fig. 3a). However, among KEGG pathways with known HIF-1A target genes, we found a significant enrichment of metabolic intermediates almost confined to TCA cycle (hypergeometric test, $q$-value $< 0.05$), suggesting that changes in HIF-1A activity alone are sufficient to affect TCA cycle. In order to validate this association, we monitored dynamic intracellular metabolic changes upon HIF-1A mRNA degradation (i.e., siRNA knockdown) in IGROV1 ovarian cancer cells, exhibiting an average basal level of HIF-1A activity (Supplementary Fig. 5). While the earliest time-points are dominated by minor and fluctuating metabolic changes, likely reflecting a general response to siRNA transfection, strong metabolic changes gradually

emerged 68 h post-transfection, and were most pronounced at 111 h after HIF-1A silencing (Fig. 3b, Supplementary Data 3). The most prominent metabolic changes (Fig. 3c) involve the accumulation of metabolites in the oxidative branch of TCA cycle (Fig. 3d), including citrate, oxalosuccinate, and N-acetyl glutamate, a downstream product of 2-oxoglutarate. The increase in abundance of TCA cycle intermediates upon HIF-1A knockdown is consistent with the previously reported regulatory role of HIF-1A as a repressor of oxidative metabolism[40] via induction of pyruvate dehydrogenase kinase (PDK), and is in agreement with the enrichment analysis of functional associations in our TR-metabolite association network (Fig. 3a). Moreover, out of 728 TRs in the TR-metabolite association network, HIF-1A was recovered within the top 2% of TRs with the strongest associations to metabolites affected by HIF-1A knockdown (Fig. 3e and Supplementary Fig. 7).

Hence, besides generating experimentally testable hypotheses on condition-specific regulatory roles of TRs in metabolism (Fig. 3a), the herein-established TR-metabolite association network might serve as a guide to extract signatures of TR deregulation directly from metabolome profiles. Such an approach would be particularly attractive for the interpretation of large collections of in vivo metabolome profiles acquired from tissue samples in patient cohort studies. In the following, we verify whether TR-metabolite associations inferred in vitro are relevant also in an in vivo context, by testing the well-known role of HIF-1A as a key oncodriver in clear-cell renal cell carcinoma.

**Predicting TRs mediating in vivo metabolic reprogramming.** Here, we ask whether the map of TR-metabolite associations found in vitro recapitulates metabolic rearrangements in an in vivo setting. To this end, we searched for TRs potentially responsible for metabolic differences between healthy and tumor tissue by evaluating the dot product between the TR-metabolite correlation matrix derived in vitro, and in vivo metabolite fold-changes between normal and cancer tissues. We applied this approach to analyze differences in metabolite abundances between clear-cell renal cell carcinoma (ccRCC) and proximal normal tissue samples in a cohort of 138 patients[7] (Supplementary Data 3). Because this independent metabolomics data set contains only a subset of metabolites (i.e., 134) detected in our TR-metabolite association network, for each patient, we estimated the significance of a TR in explaining the observed metabolic changes using a permutation test, and ranked the 728 TRs according to the median $q$-value across patients (Fig. 4a, Supplementary Data 3).

Loss-of-function mutations in van Hippel Lindau factor (VHL) gene are the most frequent and specific genetic event observed in ccRCC[41], entailing a hyper-activation of hypoxia-inducible factors (HIF-1, HIF-2, and HIF-3)[39,40]. In agreement with the genetic basis of ccRCC, we identified VHL and HIF-1A among the top 1% of TRRUST-listed[24] transcriptional regulators that potentially mediate metabolic rearrangement in ccRCC (Fig. 4a). Other top-ranking TRs include YY1 that has been shown to interact with hypoxia-inducible factors[42] (see also Supplementary Discussion and Supplementary Fig. 7). These results independently demonstrate the in vivo relevance of the previously inferred in vitro map of TR-metabolite associations, and support its potential clinical applicability to decipher metabolic rearrangements in tumor tissue samples.

To further illustrate the potential of TR-metabolite associations in aiding the interpretation of tumor-specific metabolic changes, we collected data from two additional studies monitoring tumor metabolic reprogramming in cohorts of 10 and 21 patients with colon[12] and lung cancer[43], respectively. In contrast to ccRCC,

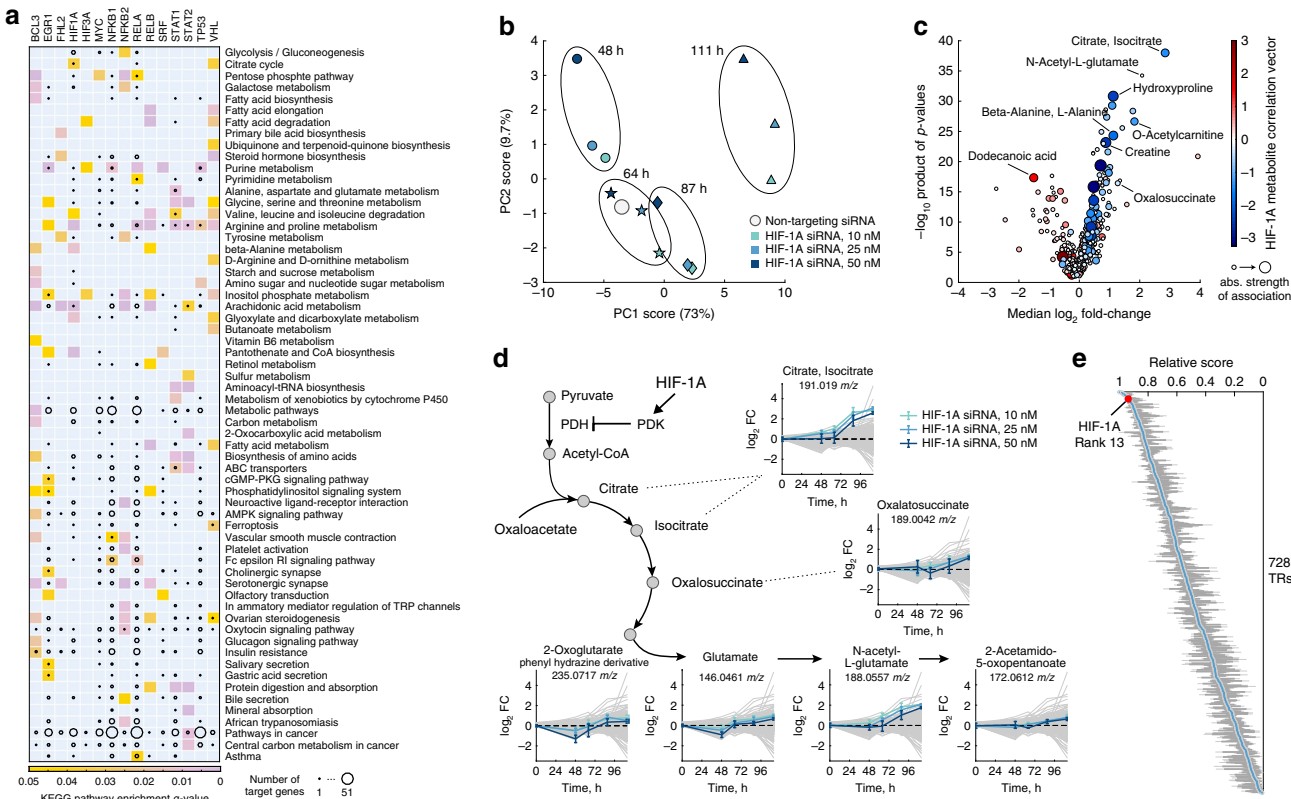

**Fig. 3** Predicted functional association of TRs to metabolic pathways. **a** Significant over-representation in KEGG metabolic pathways for metabolites associated with 15 selected cancer-related TRs[38] (as defined by the COSMIC cancer gene census). The heatmap shows metabolic pathways with a significant enrichment (hypergeometric test, q-value < 0.05). The size of black circles scales with the number of known TR-gene targets[24] in the metabolic pathway. **b** Principal component analysis of dynamic metabolome changes in IGROV1 ovarian cancer cells transfected with three different concentrations (10, 25, and 50 nM) of HIF-1A siRNA, and a non-targeting siRNA as negative control. **c** Volcano plot of metabolic changes at 111 h after HIF-1A siRNA transfection. Each putatively annotated metabolite is associated with a fold-change and significance (product of p-values across three siRNA concentrations). Only ions annotated to KEGG identifiers are shown. Dot size reports on the strength of the association predicted between HIF-1A activity and the metabolite in our TR-metabolite association network, while the color indicates the sign of the association. **d** Dynamic metabolite changes of metabolites in and proximal to TCA cycle upon siRNA transfection (with time-point 0 representing the negative control). Data are mean ± standard deviation across three replicates. Gray lines represent dynamic changes of all other detected metabolites. **e** The 728 TRs in the TR-metabolite association network were ranked by the median overlap of TR-metabolite associations with the metabolic profiles of the HIF-1A knockdown (111 h post-transfection). For each TR, we calculated a score representing the dot product of the TR-metabolite association vector and metabolite fold-changes, normalized by the sum of absolute correlations in the association network. The scores are shown as boxplots across the three siRNA concentrations (see also Supplementary Fig. 7), with blue dots indicating the median score, and edges showing 25th and 75th percentiles

the most recurrent genetic events in colon and lung cancers are mutations in p53[44,45], similarly to many different tumor types. Metabolome-based predictions suggest that deregulation of TRs other than p53, such as NF-Y or Mllt10 (Fig. 4b, c and Supplementary Discussion), more directly associates to metabolic changes observed in lung and colon tissue samples. While we don't have direct experimental evidence, the predicted TRs can function as effectors up- or downstream of p53, and unveil key tumor-specific characteristics that can be exploited in a therapeutic setting.

Consistent with this hypothesis, by analyzing the dependency between TR activity and the sensitivity of NCI-60 tumor cell lines to 130 FDA-approved drugs[31], we found 392 TRs for which activity significantly (linear regression p-value ≤ 2.67e−4, Bonferroni-adjusted threshold) associates with at least one drug sensitivity profile (Supplementary Data 4). To disentangle the difference in drug sensitivity relating to variable TR activity from those attributable to the tissue of origin, we used a multivariate statistical approach that uncovers the potential association between individual TRs and drug susceptibility. When testing these associations between groups of TRs regulating similar

cellular processes and drugs with a shared mode of action (MoA), we found expected and potentially new functional associations (Fig. 4d). For example, cell lines differentially susceptible to mTOR inhibitors that can induce cell cycle arrest[46] exhibited similar patterns in the activity of TRs involved in regulating cell cycle progression (linear regression p-value ≤ 1e−4, Bonferroni-adjusted threshold). Even stronger but less intuitive is the predicted association between regulators of calcium homeostasis and inhibitors of the MAPK signaling pathway (Fig. 4d). In the more specific cases described above, our analysis of HIF-1A and VHL activities across cell lines recapitulate the action of HIF-1A inhibitor vorinostat[47] and HIF-1A inducer imiquimod[48] (Fig. 4e, f).

Overall, by analyzing cell line responses to largely diverse drugs, we uncovered numerous TRs whose activity correlates with drug sensitivity, providing independent experimental evidence that diverse transcriptional programs can affect the survival and drug tolerance of cancer cells. Hence, similarly to gene expression signatures used clinically to guide treatment decisions[49], metabolome-based signatures of TR deregulation could open new possibilities in aiding the selection of personalized

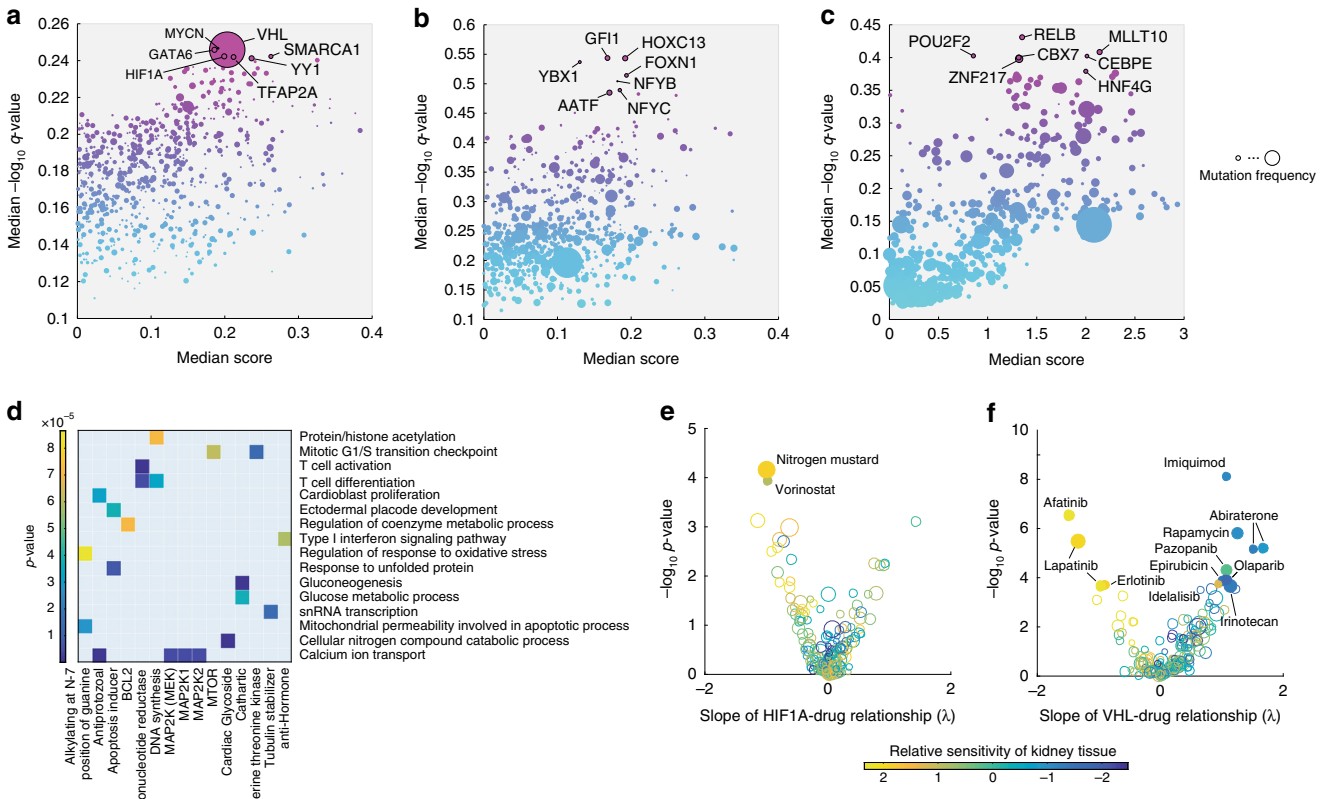

**Fig. 4** Inferring TRs as mediators of in vivo metabolic changes. **a–c** Prediction results in three previously published data sets comprising metabolic profiles of tumor vs. proximal healthy tissue from 21, 138, and 10 patients with clear-cell renal cell carcinoma[7] (**a**), lung cancer[43] (**b**), and colon cancer[12] (**c**). Each dot represents the strength (x-axis, median score) and significance (y-axis, q-value, corrected for multiple tests, blue to purple color range) of a TR in mediating in vivo metabolic changes. Dot-size indicates the frequency of mutations in the TR-encoding gene for the respective cancer type, derived from the Catalog of Somatic Mutations In Cancer (COSMIC)[38]. Gene-name labels are shown for the top 1% most significant hits (see Supplementary Discussion and Supplementary Fig. 7). **d** 131 clusters of functionally related TRs (Supplementary Data 3) were tested against 82 different drug modes of action (MoA) to find a significant over-representation of drug-TR associations. Colored squares indicate the significance of the enrichment analysis. Only TR clusters and drug MoAs with at least one significant association (p-value < 1e−4, Bonferroni-adjusted threshold) are shown in the heatmap. **e**, **f** Each dot represents one of the tested FDA-approved drugs. Estimated influence of changes in TR activity on drug susceptibility (i.e., λ) and corresponding p-value significance are reported. Dot size is proportional to the number of TRs that exhibit a significant association (linear regression p-value ≤ 2.67e−4, Bonferroni-adjusted threshold) with the drug, while dot color reflects the estimated average susceptibility of renal cell lines (i.e., $\beta_{kidney}$) with respect to the remaining seven tissue types

therapeutic treatments. Taken together, in vitro associations between TRs and metabolites not only allow predicting TR regulatory functions in metabolism, but could also complement genomic information and guide the analysis, molecular classification and interpretation of metabolome profiles from large patient cohorts.

**Systematic prediction of potential modulators of TR activity.** While we have shown that metabolic rearrangements in cancer can be correlated to changes in TR activity, the origin of such changes often remains elusive. Mutations in genes encoding transcriptional regulators can be directly responsible for altered TR functionality, and in some cases even explain disease etiology[50]. However, the activation of new transcriptional programs is often an indirect response to changes in the abundance of internal effectors of cell signaling[51,52]. In silico models have proven extremely powerful in finding new allosteric interactions that can regulate enzyme activity[53] and in testing their in vivo functionality[54], but little progress has been made in the systematic mapping of effectors of TR activity. Here, we integrated three layers of biological information—i.e., metabolome, proteome, and transcriptome, to obtain first insights into how cells can activate

distinct transcriptional programs in response to changes in the abundance of specific intracellular metabolites.

Analysis of phosphorylation interaction networks in yeast[55] and human[56] revealed that TRs, as compared to enzymes, interact on average with many more kinases (Kolmogorov–Smirnov test, p-value ≤ 0.001, Fig. 5a, b), reflecting a key role of TRs as mediators in phosphorylation signaling cascades and emphasizing the importance of elucidating post-translational modulators of TR activity. While the impact of phosphorylation has been systematically studied[55,56], much less is known about the potential role of metabolites in the allosteric regulation of TR activity. Despite the increased interest in metabolites as signaling molecules and their potential role in driving cellular transformation[51], resolving the influence of metabolites on TR activity has remained a daunting task. Here, we established an in silico framework for generating hypotheses on regulatory interactions between TRs, metabolites and kinases (Fig. 5c). To that end, we used model-based fitting analysis to integrate TR activity and metabolome profiles with proteome data[11] measuring the abundance of 100 TRs and 64 kinases/phosphatases across 53 cell lines. For each TR, we applied non-linear regression analysis to determine whether variation in TR activity could be modeled as a function of TR protein

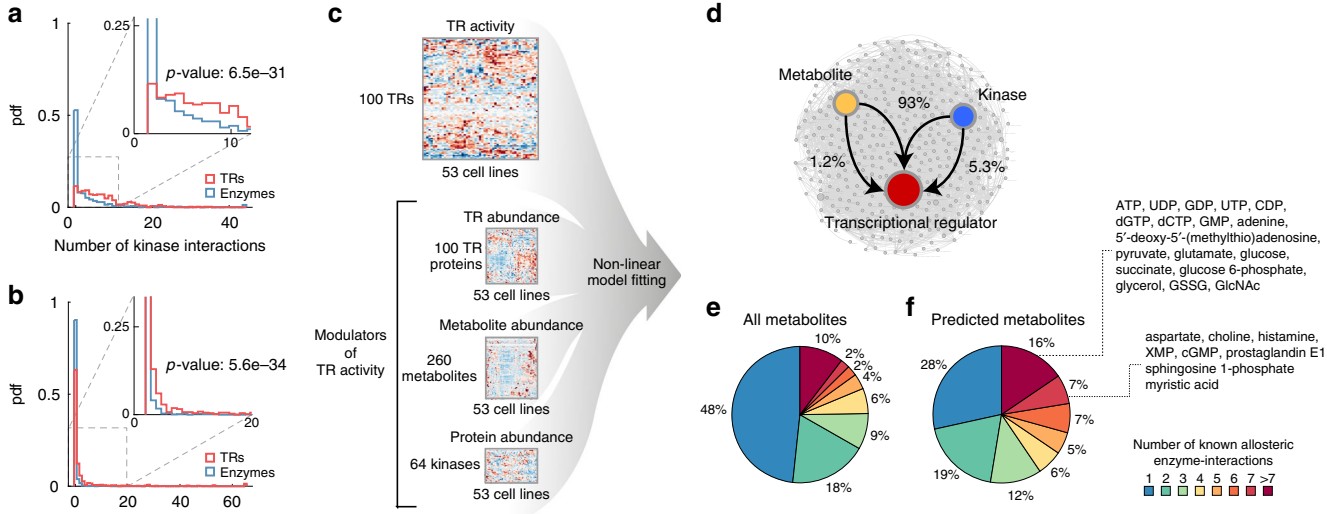

**Fig. 5** Modeling TR activity using an ensemble of non-linear models. **a**, **b** Probability density function (pdf) of the number of kinases reported to interact with TRs (red) and metabolic enzymes (blue) in yeast[55] (**a**) and human[56] (**b**). *P*-values report on the significance of the difference between TRs and enzyme pdfs (Kolmogorov–Smirnov test). **c** Schematic overview of the computational framework. Three layers of biological information are integrated using a model-based fitting approach: protein abundance of 100 TRs and 64 kinases, respective TR activities derived from transcriptome data using NCA and relative levels of 260 metabolites across cell lines (see Methods section). **d** Predicted network of modulatory interactions between TRs, metabolites and kinases. Most interactions involve the combined action of a metabolite and a kinase (93% of interactions, Supplementary Data 5). **e**, **f** The pie chart represents the distribution of metabolites allosterically regulating one or more enzymatic reactions. All metabolites (1633) with at least one known allosteric interaction in human[26] are reported in **e**, while only metabolites predicted to modulate the activity of at least one TR are shown in panel **f**. Metabolites that regulate more than seven enzymatic reaction are lumped together

abundance and the activating or inhibiting action of individual metabolites and/or kinases. In total, we tested 6,753,600 models, and found 1888 interactions which significantly (FDR ≤ 0.1%) improved the explained variance in the activity of 96 TRs out of the 100 TRs for which protein abundance data was available[11]. Consistent with the dominant role of phosphorylation in modulating TR activity (Fig. 5a, b), we found that on average TRs are predicted to interact with more kinases than metabolites (Supplementary Fig. 8). However, most of the inferred regulatory interactions (93%) involved the combined action of a kinase and a metabolite, while only few metabolites or kinases could alone significantly improve model-fitting (Fig. 5d, Supplementary Data 5). This observation suggests that multiple coordinated regulatory mechanisms possibly underlie the post-transcriptional regulation of TR activity. Moreover, metabolites predicted to affect TR activity are significantly enriched (hypergeometric test, *p*-value 3.1e−4, Supplementary Fig. 8) for key signaling molecules in the cell known to allosterically regulate multiple enzymatic reactions, such as glutathione, glutamate or ATP (Fig. 5e, f). Among the most highly connected metabolites predicted by our model-based fitting analysis is choline (118 interactions with 36 TRs), in agreement with recent studies showing that altered choline levels are a metabolic hallmark of malignant transformation[57] (see Supplementary Discussion).

Overall, our approach opens the door for a systematic investigation of a previously largely unexplored[58] interaction space between transcriptional regulators and signaling effectors in human cells. An extensive network of post-translational regulatory interactions has emerged in model organisms such as *E. coli*[26] or yeast[55,59], and based on our findings we expect a similar picture to hold true in human cells.

## Discussion

In recent years, increasing efforts have been made to understand the signals driving metabolic changes in cancer[12]. Transcriptional regulation is at the basis of the decision-making process of a cell and its ability to allocate resources necessary for cell transformation and proliferation. Genome sequencing and transcriptome technologies have revealed an intricate network of TR-gene regulatory interactions in which mutations in TRs often associate to disease states and aberrant metabolic phenotypes[7,60]. However, it is important to emphasize that gene regulatory interactions between enzymes and TRs per se are not sufficient to functionally regulate the activity of metabolic pathways. Key to unambiguously resolving regulatory circuits at the intersection with metabolism are methods searching for coordinated behavior between the different levels. To this end, we developed a combined experimental and computational framework that overcomes important limitations in large-scale metabolome screenings, including (i) the limited throughput and laborious sample preparation of classical metabolomics approaches in mammalian cell cultures, (ii) the lack of scalable methods to adequately normalize metabolomics data across morphologically diverse cell types, and (iii) the need for systematic data integration strategies. The herein-proposed workflow for large-scale metabolome profiling is directly applicable to the study of dynamic metabolic responses to external stimuli[18], and can scale to larger cohorts that are now within reach of other molecular profiling platforms[61].

By combining cross-sectional omics data from diverse tumor cell lines, we constructed a global network model across three layers of biological information: the transcriptome, the proteome, and the metabolome, exploiting the naturally occurring phenotypic diversity in an in vitro cell line system. By analyzing the coordinated changes in baseline transcriptome, proteome and metabolome with the aid of a gene regulatory network and model-based fitting analysis, we investigated the bi-directional exchange of signaling information between TRs and metabolic pathways. For many TRs, we predict potentially new regulatory associations with central metabolic pathways, suggesting a large space of transcriptional solutions by which cells can fulfill the anabolic and catabolic requirements for rapid proliferation and

adaptation to nutrient limitations. The correlation between glycolytic flux and activity of TRs involved in the response to nutrient limitation and stress hints at the ability of cancer cells to sidetrack transcriptional programs activated by nutrient scarcity or stresses to potentially fulfill carbon demand for fast proliferation. Moreover, we observed a global coordination between glucose and one-carbon metabolism, which revealed a selective sensitivity to antifolate drugs in cell lines with low glucose uptake and might serve as a diagnostic marker for cancer cells that are more likely to respond to folate synthesis inhibitors.

Because measuring intracellular fluxes is still a major challenge, and these measurements are typically limited to central metabolic pathways, our metabolome profiling technique offers an alternative tool to probe metabolic regulation at a genome-scale and high-throughput in cancer cells. In light of the central regulatory role of TRs in cellular organization, targeting transcriptional regulators is an extremely attractive way to counteract global gene expression changes that underlie cancer survival and development[62,63]. Endogenous metabolites capable of modulating TR activity could become invaluable chemical scaffolds to design new therapeutic molecules targeting oncogenic TRs, with the potential to overcome difficulties related to targeting kinase-mediated signaling cascades[63]. Altogether, our work also suggests that, while clearly far from typical in vivo conditions, in vitro cell line systems represent an invaluable discovery tool to investigate metabolic regulatory mechanisms that can still generalize to in vivo conditions and clinical settings. The experimental and computational framework proposed in this study is applicable to other systems or diseases, providing us with an unprecedented tool to investigate the origin of metabolic dysregulation in human diseases.

## Methods

**Cell cultivation**. The NCI-60 cancer cell lines were obtained from the National Cancer Institute (NCI, Bethesda, MD, USA). The full panel consists of 60 cell lines, 54 of which grow adherently and were used in this study. Of note, cell line MDA-N had previously been excluded from the panel, but was replaced by the NCI recently by MDA-MB-468 cells. After thawing, the 54 adherent cell lines were expanded in cell culture flasks (Nunc T75, Thermo Scientific) at 37 °C and 5% $CO_2$ in RPMI-1640 (Biological Industries, cat.no. 01-101-1A) supplemented with 5% fetal bovine serum (FBS, Sigma Aldrich, F6178), 2 mM L-glutamine (Gibco, cat.no. 25030024), 2 g/L D-glucose (Sigma Aldrich, cat.no. G8644), and 100 U/mL penicillin/streptomycin (P/S, Gibco, cat.no. 15140122). After two passages, the cells were transferred to fresh medium where FBS was replaced by dialyzed FBS (dFBS, Sigma Aldrich, cat.no. F0392) with a reduced content of low molecular weight compounds, to improve the accuracy of metabolite quantification. Twenty cell lines were exemplarily tested and confirmed mycoplasma-free: SF268, IGROV1, UACC62, HCC2998, DU145, NCI-ADRRESS, OVCAR3, SNB19, SKMEL18, NCI-H23, SF539, HOP62, UACC157, NCI-H322M, NCI-H522, OVCAR4, EKVX, UO31, CAKI-1, MDAMB231.

Cells were maintained in medium with dFBS throughout all experiments. The starting cell density for metabolomics experiments in 96-well plates (Nunc cat.no. 167008, Thermo Scientific) was determined for each cell line. To this end, cells were plated in triplicates at eight different starting cell densities and incubated at 37 °C and 5% $CO_2$ for 3 days. On the third day, the medium was changed in all wells by aspirating the spent medium using a multichannel aspirator, washing once with phosphate-buffered saline solution (PBS, pH 7.4, 37 °C, Gibco, cat.no. 10010015) using a multichannel dispensing pipet, and finally filling each well again with 150 µL of fresh medium. The plate is imaged to determine the confluence (see below) immediately before and after media change, and after 72 h. The starting cell density for metabolomics experiments was then chosen to guarantee a minimum of 20-30% cell confluence after media change, and approx. 80% confluence after 72 h.

**Cell imaging and image analysis**. We monitor cell growth by measuring cell confluence (i.e., area of the well covered by cells in percentage) directly from 96-well plates (Nunc cat.no. 167008, Thermo Scientific) using automated time-lapse microscopy imaging. Every 1.5 h, bright-field microscopy images of each well were acquired using a TECAN Spark 10 M plate reader. In addition, we developed an image analysis framework to segment cells and determine the characteristic cell size area (i.e., average surface area of single adherent cells) for each cell line (Supplementary Fig. 1 and Supplementary Note). To quantify the number of extracted cells, confluence was divided by the characteristic cell size (Supplementary Fig. 1).

A detailed description and validation of the algorithm used for estimating cell numbers from bright-field microscopy images is provided in the Supplementary Note, alongside with a Matlab code. Of note, this approach has several important advantages, in that it is non-destructive, and allows quantifying cell growth and cell numbers without any manual sample manipulation.

**Metabolomics experiments**. Cells were plated in triplicates in 150 µL of RPMI-1640 medium (5% dFBS, 2 g/L glucose, 2 mM glutamine, 1% P/S) in 96-microtiter well plates. After an initial growth phase, the medium in each well was renewed on the third day, and the cultures were subsequently monitored for four more days (96 h). Ten replicate plates were prepared in each experiment to allow for generating metabolomics samples at five different time-points (immediately before media change, and at 24, 48, 72 and 96 h after media change), and one additional plate for continuous growth monitoring (TECAN Spark 10 M, 37 °C, 5% $CO_2$).

At each sampling time point, two replicate 96-well plates were processed (plates A and B). Plate A was used to generate cell extracts, by (1) removing the spent medium, (2) washing once with 75 mM ammonium carbonate (pH 7.4, 37 °C), and (3) adding ice-cold extraction solvent (40% methanol, 40% acetonitrile, 20% water, 25 µM phenyl hydrazine[64]). Finally, the plate is sealed, incubated at −20 °C for one hour, and subsequently stored at -80 °C until MS analysis. Plate B undergoes the same processing steps, except for the last one, where each well is filled with PBS (pH 7.4, 37 °C), and the plate is immediately imaged to determine cell confluence for subsequent normalization of MS spectra.

Immediately prior to MS analysis, the plates were thawed on ice, and the extracted cells were scraped off the bottom of each well using a multichannel pipet with wide-bore tips. Next, the cell extracts were transferred to 96-well plates with conical bottom and centrifuged at 4 °C, 4000 rpm for 5 min to separate cell debris. Finally, pooled cell extracts for each experiment (pooled from five cell lines processed within the same experiment) as well as aliquots of cell-free extraction solvent were added to each measurement plate as control samples, and the plates were sealed and stored at 4 °C until injection.

**Metabolome profiling using FIA-TOFMS**. Cell extract samples were analyzed by flow-injection analysis time-of-flight mass spectrometry (FIA-TOFMS) on an Agilent 6550 iFunnel Q-TOF LC-MS System (Agilent Technologies, Santa Clara, CA, USA), as described by Fuhrer et al.[14]. This method allows generating high-resolution spectral profiles in less than one minute per sample, allowing for sensitive high-throughput profiling of large sample collections. In brief, a defined sample volume of 5 µL is injected using a Gerstel MPS2 autosampler into a constant flow of isopropanol/water (60:40, v/v) buffered with 5 mM ammonium carbonate (pH 9), containing two compounds for online mass axis correction: 3-Amino-1-propanesulfonic acid, (138.0230374 m/z, Sigma Aldrich, cat. no. A76109) and hexakis(1H,1H,3H-tetrafluoropropoxy)phosphazine (940.0003763 m/z, HP-0921, Agilent Technologies, Santa Clara, CA, USA). The sample plug is delivered directly to the ion source for ionization in negative mode (325 °C source temperature, 5 L/min drying gas, 30 psig nebulizer pressure, 175 V fragmentor voltage, 65 V skimmer voltage). Mass spectra were recorded in the mass range 50–1000 m/z in 4 GHz high-resolution mode with an acquisition rate of 1.4 spectra per second. Raw MS profiles were processed to align spectra and pick centroid ion masses using an in-house data processing environment in Matlab R2015b (The Mathworks, Natick).

**Multiple hypothesis testing correction**. Given a sufficient number of tests, the Storey method[65] for correction of multiple hypothesis testing was adopted (i.e., q-value). However, this methodology typically requires a large number of null tests to derive an accurate estimate of $\pi_0$ (estimate of the proportion of true null p-values)[65]. In cases where the number of tests is inadequate for q-value correction, we adopted the Bonferroni correction. Only in two cases, we applied a resampling confidence interval-based method: (i) to generate Fig. 2c of strongest TR-metabolite associations, and (ii) to select the strongest TR-metabolite-kinase predicted interactions (Fig. 5d). In these cases, to avoid any a priori assumptions of the underlying distributions, we determine the false discovery rate (FDR) from the bulk distribution of correlation or mean squared error values, respectively, derived from random sampling. The procedure is described in details in the respective sections.

**Metabolite annotation**. Measured ions were putatively annotated by matching mass-to-charge ratios to a reference list of calculated masses of metabolites listed in the Human Metabolome Database (HMDB) and the genome-scale reconstruction of human metabolism[21] (Recon2) within 0.003 amu mass accuracy. The reference mass list was generated from the respective sum formulae, considering deprotonation as the most prevalent mode of ionization in the chosen acquisition conditions. To allow for the annotation of α-keto acid derivatives formed in presence of phenyl hydrazine in the extraction solvent[64], sum formulae for the phenylhydrazone derivatives ($+C_6H_8N_2-H_2O$) of a total of 30 α-keto acid compounds (selected via KEGG SimComp search http://www.genome.jp/tools/simcomp/) were added to the metabolite list for annotation. The final list of putatively annotated metabolites consisted of 689 and 5949 unique compound IDs from Recon2, and HMDB, respectively.

**Data normalization**. We corrected for systematic errors using a two-step regression model to disentangle the contributions of extracted cell numbers, plate-to-plate variance, instrumental and background noise from the actual variance in metabolite abundances between cell types (Supplementary Fig. 2). To this end, raw data were first corrected for instrument drift by normalizing for possible batch/plate effects. Each plate contains 12 pooled cell extract samples prepared from the 5 different cell lines in each experiment (i.e., batch). Measured intensities for each annotated ion are modeled as follows:

$$I_{i,j,p} = \gamma_p \cdot M_{i,j} \tag{1}$$

$$\log\left(I_{i,j,p}\right) = \log(\gamma_p) + \log(M_{i,j}) \tag{2}$$

Where $I_{i,j,p}$ is the measured intensity for ion I, in pooled sample j and plate p, $\gamma_p$ is the scaling factor associated to each plate and $M_{i,j}$ represents the actual abundance of metabolite i in sample j. By using a linear regression scheme we can estimate both parameters ($\gamma_p$ and $M_{i,j}$) within an unknown scaling factor. After correcting for possible instrumental artifacts, we implemented a second step in order to derive comparative measurements of metabolite abundance for each cell line. Here, we follow each cell line along the linear growth phase, sampling every 24 h across 5 days. We typically obtain 15 data points for each cell line at different cell densities. The expectation is that the signal measured for any ion of biological origin (i.e., a genuine metabolite) would increase linearly as the number of cells in the sample increases. The proportionality (i.e., α parameter) between ion intensity and extracted cells depend on the intracellular concentration of the metabolite. Here, by implementing a multiple regression scheme, we estimate the relative abundance of a given metabolite in each of the cell lines, α, alongside with its standard error (Fig. 1b, Supplementary Fig. 2). A linear regression model describes variation in ion intensity as a linear function of cells extracted (α) and a constant parameter (β) that capture MS background noise. For each ion, the α values are specific of each cell line while the constant term in the model is fixed (i.e., expected ion signal at zero confluence that is independent of the cell line).

Because of the large number of measurements for each cell line at different cell densities, we can apply a multiple regression analysis (fitlm function in Matlab) to infer all model parameters αs and β at once, by minimizing the Euclidian distance between measured metabolite intensities and model predictions. For each metabolite, we solve the following linear model, including all 54 cell lines:

$$
\begin{bmatrix}
I_{cell_1,1} \\
I_{cell_1,2} \\
I_{cell_1,3} \\
\cdots \\
I_{cell_2,1} \\
I_{cell_2,2} \\
I_{cell_2,3} \\
\cdots \\
I_{cell_c,s}
\end{bmatrix}
=
\begin{bmatrix}
N_{cell_1,1} & 0 & \cdots & 0 & 1 \\
N_{cell_1,2} & 0 & \cdots & 0 & 1 \\
N_{cell_1,3} & 0 & \cdots & 0 & 1 \\
\cdots & \cdots & \cdots & \cdots & \cdots \\
0 & N_{cell_2,1} & \cdots & 0 & 1 \\
0 & N_{cell_2,2} & \cdots & 0 & 1 \\
0 & N_{cell_2,3} & \cdots & 0 & 1 \\
\cdots & \cdots & \cdots & \cdots & \cdots \\
0 & 0 & \cdots & N_{cell_c,s} & 1
\end{bmatrix}
\cdot
\begin{bmatrix}
\alpha_{cell_1} & \alpha_{cell_2} & \cdots & \alpha_{cell_c} & \beta
\end{bmatrix}
\tag{3}
$$

Where $I_{cell\ c,s}$ is the measured metabolite intensity in sample s of cell line c, $N_{cell\ c,s}$ is the number of cells extracted in sample s of cell line c, and αs (for each cell line) and β are the unknown parameters to be fitted. The number of cells per sample is derived from the confluence measurements at sampling, divided by the average cell size area determined using our image segmentation analysis (see Supplementary Fig. 2 and Supplementary Note). After this step, we retained 2181 ions with a regression p-value below a threshold value of 3.4e-7 (adjusted by the number of cell lines and ions) in at least one cell line, and that showed a significant dependency with the extracted cell number in more than 80% of cell lines (Supplementary Fig. 2). Of note, we found that prior to normalization, the variance across three biological replicates at the same time-point was equally low in cell confluence (median: 7.4% CV) and raw ion intensities (median: 13%, Supplementary Fig. 2), reflecting the high quality of MS measurements.

In the third and last step, we take into account systematic changes in metabolite abundances related to differences in cell size (i.e., cell volume) between the 54 cell lines to derive comparative estimates of intracellular metabolite concentration. Principal component analysis of relative metabolite abundance per cell revealed a strong trend across the 54 cell lines (PC1, 58.9% explained variance, Supplementary Fig. 2), which strongly correlates with cell line volumes derived from cell diameters measured in ref. [66] as well as with the herein determined adherent cell size area (Supplementary Fig. 2, see Supplementary Note). The transitive correlation between adherent cell size and the spherical cell volume in suspension indicates that adherent cell height can be approximated as a constant. To correct MS data for differences in cell line volumes, we selected 987 ions that showed a significant and strong correlation (Pearson's $r > 0.8$, $p < 0.05$, Supplementary Fig. 2) with PC1. KEGG pathway enrichment analysis showed that these putatively annotated metabolites were strongly overrepresented in fatty acid metabolism (Supplementary Fig. 2), consistent with the expected linear dependency between cell membrane surface (i.e., phospholipid content) and cell volume of adherent cells. For each ion, cell line-specific α-values of the selected metabolites across all 54

cell lines were used to calculate a consensus correction factor for each cell line by taking the mean across the 987 ions. To apply the cell volume correction to the full data set, we divided the cell line-specific α-values for each ion by the consensus correction factor (Supplementary Data 1).

Finally, the corrected α-values were normalized using Z-score normalization:

$$Z_\alpha = \frac{\alpha_{cell} - \bar{\alpha}}{\sqrt{\frac{1}{n}\sum_{cell=1}^{n}(\alpha_{cell} - \bar{\alpha})}}, \tag{4}$$

where $n$ is the number of cell lines (i.e., 54).

The final normalized data set is provided in Supplementary Data 1 alongside with p-values and standard errors derived from regression analysis. Missing values (NaN) correspond to cases where the measured ion abundance for the annotated metabolite was close to the background level in the cell line, and can be considered as zero for further analysis. In cell lines where a significant dependency (p-value < 3.4e−7, Bonferroni-adjusted threshold) of given metabolite's abundance with cell number could be robustly determined and exceeded the background noise, relative standard errors of α calculated during fitting analysis were below 20% (median: 11%, Supplementary Fig. 2).

**Analysis of tissue signatures across four omics data types**. A complete description of the analysis can be found in Supplementary Methods.

**Estimating TR activity by network component analysis (NCA)**. Originally established by Liao et al.[29], NCA provides a mathematical framework for reconstructing TR regulatory signals (TR activity) from gene expression profiles. Here, we adopted sparseNCA implementation by Noor et al.[67] (Matlab code downloaded from https://sites.google.com/site/aminanoor/softwares). This methodology adopts a mathematical model to approximate TR-target regulatory interactions and integrates prior network information with the expression of target genes across multiple conditions to regress the activity of the respective TRs, delivering a relative measure of TR activity. We obtained normalized gene expression profiles across the NCI-60 cell lines from Gene Expression Omnibus (accession number GSE32474), containing 54,675 mRNA probes. TRRUST database[24] served as the source of TR-target gene interactions relevant in human, including 748 human TRs and 1975 non-TR gene targets. Intersecting these two resources, we assembled a network of 2209 unique genes corresponding to 5490 mRNA probes that match target genes of 728 TRs in the TRRUST database (Supplementary Fig. 5). We implemented a bootstrapping approach to account for incompleteness of the regulatory network (i.e., missing regulatory interactions), and the fact that there may be multiple optima in the solution space. Of note, even if the current knowledge of TR-target genes is incomplete, few gene targets can be sufficient to estimate TR relative activities using this approach. To this end, for each TR we randomly selected 48 additional TRs and constructed a sub-network containing the 49 TRs and their target genes. Because growth-rate has a pleiotropic effect on gene expression, here reflected in the correlation between first principal component of gene expression data and cell line growth rates (Supplementary Fig. 5), we decouple TR activity from the confounding effect of growth-rate by adding an additional TR that targets all genes. This fictitious TR mimics the general effect of proliferation rates on transcription. As a result, each TR is embedded in a sub-network of 50 TRs and their target genes from the full network. Ten such subnetworks were created randomly for each TR to apply NCA. In this bootstrapping scheme, each TR was sampled in on average 490 subnetworks (permutations, min. 423, max. 556 data points per TR). In the final data set, we normalized the calculated TR activity to the maximum across all permutations, and finally calculated the median TR activity and its standard deviation for each TR and cell line (Supplementary Fig. 5). It is worth noting that the estimates we obtain with this approach are correct within an unknown scaling factor, and hence we determine a relative measure of activity for each of the 728 TRs across the NCI-60 cell lines (Supplementary Data 2).

**TR-metabolite association network**. In order to find metabolites whose relative abundances correlate with TR activity, we calculated pairwise Spearman correlations between all 2181 annotated metabolites and 728 TRs across the 54 cell lines (Supplementary Data 3).

For visualization in Fig. 2c, we controlled the false discovery rate (FDR) among network links at 0.1% using a bootstrapping approach to calculate the 99.9% confidence interval of correlation coefficients after randomizing the data set. To that end, the cell lines in the metabolome data set were randomized by resampling 100 times with replacement, and pairwise Spearman correlation coefficients were calculated for each randomized data set. Correlation coefficients yielding a 99.9% confidence interval (0.1% FDR) were obtained from the pooled list of absolute correlation coefficients by finding the smallest correlation coefficient that exceeds the maximum value among 99.9% lowest correlation coefficients).

**Measurement of glucose and lactate exchange rates**. A complete description of the experiments and techniques used to determine glucose uptake- and lactate secretion rates can be found in Supplementary Methods.

**Pathway enrichment in TR-metabolite correlation signatures**. To assess the over-representation of TR-metabolite associations in KEGG metabolic pathways, the pairwise correlations between TR activity and metabolite relative abundance across cell lines were rank-transformed. A statistical score that models the probability of a KEGG pathway to be significantly associated to a TR is based on the collective activities of multiple metabolites in a pathway following the approach described in ref. [68]. The significance of the rank distribution of all metabolites within the same KEGG pathway is tested by means of an iterative hypergeometric test, indicating the statistical significance of metabolic intermediates of a common metabolic pathway (e.g., TCA cycle) being distributed toward the top ranking ones. $P$-values were corrected for multiple tests by $q$-value estimation[69] (Supplementary Data 3).

**siRNA transfection and HIF-1A knockdown**. A complete description of the experimental procedures used to knockdown the TR HIF-1A in IGROV-1 ovarian cancer cells, and quantify metabolic changes in response to the knockdown can be found in Supplementary Methods.

**Inferring TR involvement from in vivo metabolic changes**. Based on the TR-metabolite association network, the procedure used to estimate the contribution of each of the 728 TRs in mediating metabolic changes observed in vivo, consists of 3 main steps: (i) For each patient $\log_2$ fold-changes of detected metabolites between cancer and adjacent normal tissue are estimated ($\mathbf{FC}^P$), (ii) The dot product between metabolite fold-changes and the TR-metabolite correlation vector ($\mathbf{c}_{TR}$, product of correlation R and $-\log_{10}$ p) estimated from in vitro cell lines is computed for each TR ($S_{TR}^P$), (iii) the significance is estimated using a permutation test, where metabolite order is shuffled 10.000 times, the dot product is estimated for each random permutation ($\tilde{S}_{TR}^P$) and $p$-values are estimated as follows:

$$S_{TR}^P = \frac{\mathbf{C}_{TR} \cdot \mathbf{FC}^P}{|| \mathbf{C}_{TR} ||^1} \qquad (5)$$

$$p-\text{value}_{TR} = \frac{\sum_k^{10,000}(\tilde{S}_{TR}^P \geq S_{TR}^P)}{10,000} \qquad (6)$$

$P$-values are corrected for multiple tests by $q$-value estimation[69], and the median across patients is calculated (Supplementary Data 3). Notably, when analyzing the data published in ref. [7] we excluded all detected metabolites with more than 10 missing values across patient samples.

**Associations between TR activity and drug action**. A complete description of the linear regression model used to quantify associations between variation in drug sensitivity and TR activity can be found in Supplementary Methods.

**Prediction of metabolite-TR effectors**. A complete description of the non-linear model fitting analysis for the prediction of metabolite- or kinase effectors of TRs can be found in Supplementary Methods.

## Data availability
All data generated or analyzed during this study are included in this published article as Supplementary Data.

## Code availability
A detailed description of all data analysis steps is published in this article in the Methods section and Supplementary Information. Matlab code for the image analysis software is available for download at http://www.imsb.ethz.ch/research/zampieri-group/resources.html.

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

## Acknowledgements

We thank the National Cancer Institute (NCI) for providing the cancer cell lines. This work was supported by a Worldwide Cancer Research (WCR-15-1058) project funding to M.Z., K.O. was funded by the Austrian Science Fund (FWF): FWF P26603 and FWF W1224 Doctoral Program BioToP—Biomolecular Technology of Proteins. We thank Uwe Sauer and Nicola Zamboni for supporting this work and providing laboratory facilities, Dimitris Christodoulou, Maren Diether, Nicola Zamboni and Victor Chubukov for helpful feedback and discussions.

## Author contributions

M.Z. designed the project. K.O. and S.D. performed the metabolome experiments. K.O. and M.Z. analysed the data. M.Z. and S.D. designed and implemented the image analysis framework. M.Z. and K.O. wrote and revised the manuscript. All authors contributed to preparing the manuscript.

## Additional information

**Competing interests:** The authors declare no competing interests.

