## [Peer Review File · Nature Communications]

This manuscript has been previously reviewed at another journal that is not operating a transparent peer review scheme. This document only contains reviewer comments and rebuttal letters for versions considered at Nature Communications. Mentions of the other journal have been redacted.

Reponse to reviewers

Reviewers' comments:

Reviewer #1:

Remarks to the Author:

“Charting the cross-functional map between transcriptional regulators and cancer metabolism” by Ortmayr et al. presented a state-of-the-art metabolomics platform performed on NCI60 cell lines at multiple time points with normalization on cell numbers and sizes. The authors also attempted to link TRs to metabolic phenotypes, which is of high potential impact. However, there are a few major points should be addressed.

We thank the reviewer for the overall positive and encouraging comments.

(1) As the authors stated “Here, we chart a genome-scale map of TR-metabolite associations in human cells using a new combined computational-experimental framework for large-scale metabolic profiling of adherent cell lines, and the integration of newly generated intracellular metabolic profiles of 54 cancer cell lines with transcriptomic and proteomic data.” This reviewer was not impressed by the late integration of proteomics data into the dataset in Figure 5. “To that end, we used model-based fitting analysis to integrate TR activity and metabolome profiles with proteome data measuring the abundance of 100 TRs and 64 kinases/phosphatases across 53 cell lines” should include metabolic enzymes.

To include enzymes in the proposed analysis we would need to be able to measure “enzyme activity”, i.e. intracellular fluxes. We are not aware of any fluxomic dataset experimentally measuring intracellular fluxes at a genome-scale across NCI60 cell lines.

All these critical state-of-the art proteome data by Guo et al. should be maximized and integrated with Figure 2 to come up with a highly-integrated, validated poly-omics TR-metabolic dataset to hammer out the discrepancies among transcriptomes, proteomes, and metabolomes for the subsequent interrogation in functional figures (current figures 3 and 4).

It is unclear what the reviewer means by “a highly-integrated, validated poly-omics TR-metabolic dataset” and more specifically what the layout and specific added value of his/her proposed dataset would be. Our work is one of the very few that uses non-linear modelling to integrate transcriptome, proteome and metabolome in a systematic framework to investigate the crosstalk between changes in metabolite levels and TR-activity.

(2) Figure 5 e and f were not called out correctly in the text.

We thank the reviewer for pointing out the incomplete Figure reference, which we have now corrected.

(3) Please remove APC and VHL from the TR list. They are not transcriptional regulators “TRs, such as transcription factors or chromatin modifiers”.

In the manuscript we adopted the inclusion criteria of the manually curated and well-established TRRUST database of gene regulatory interactions, which includes TR co-regulators, e.g. VHL. We have now again refined the definition of TR throughout the manuscript to clarify our inclusion criteria.

(4) The attempt for drug sensitivity predication as highlighted in Figure 4 e and f are not informative, especially on the VHL part. This should be reanalyzed utilizing the fully-integrated dataset as mentioned in comment (1). The CCLE dataset should be included in refining the drug treatment results. The overemphasis on drug treatment outcome on NCI-60 can easily lead to wrong statement, e.g. pazopanib inhibits directly tumor blood vessels and thereby indirectly kills tumor cells.

We revised the text to avoid any overstatement and toned down this section. Moreover, both the NCI-60 and the CCLE are cell line-based panels, hence the suggested analysis won't add neither change our conclusions.

Reviewer #2:

Remarks to the Author:

I have read the manuscript again, and I appreciate the efforts of the authors to write it better. I agree a huge amount of data analysis has been done. Nonetheless, my principal impression has not changed. The author generate a valuable dataset, and work hard to connect transcriptome with metabolome.

We thank the reviewer for her/his honest comments.

In difference to other studies that have been recently been published in this area, it does however not exceed this descriptive level.

To date, metabolome studies comparing largely diverse cell lines are extremely rare, possibly because the difficulties in comparing metabolome profiles across different cell types (e.g. different morphology, size) have been underestimated and never systematically investigated. Hence, the data and methodology described in our work represent a unique resource for the scientific community working on systems biology and cancer metabolism, to address questions beyond those specific of our study.

The network is generated, but it is not proven to be causal, or predictive for new cell lines that have not been analysed. No clear no new biological message is presented. In addition these are metabolomes for cell lines, and as metabolism is changed significantly in cell culture, its not clear how much of the network replicates in vivo. So one is left a little bit unsatisfied.

We disagree with the reviewer. We generated a unique genome-scale map of associations between TR and metabolites, and never claim causality of these associations. Moreover, we demonstrate that this map can provide invaluable information to resolve metabolic functionality of TRs, find cross-talk between metabolic pathways that can be functional to predict drug sensitivity (e.g. glucose/folate metabolism) and to predict deregulated TRs directly from metabolome profiles, even from in-vivo patient samples.

Reviewer #3:

Remarks to the Author:

I have now read the revised version of the manuscript. My primary previous concern was the lack of validation of the claims regarding transcription regulatory activity. In this regard, the authors present three new results.

- Proximity: The proximity analysis added by the authors (new Supplementary Figure 4, panel I) does show a clear trend, which is encouraging. This analysis would indicate that metabolite changes can be associated with enzymes that are regulated by TRs, and thus their levels may be indicative of TR activity. However, it is worth noting that this type of validation is indirect by nature, and the strength of the association is not particularly impressive, given that the distance appears to go from 5.4 from random expectation to 4.1. Figure 1g seems to show a much stronger proximity relationship between just the raw gene expression and metabolite data. Thus, this analysis does offer some support for their predictions but it is not convincing that their TR activity predictions are accurate from this alone.

- Glucose/lactate data: The addition of glucose and lactate exchange data is unexpected but welcome, especially if it shows significant differences with the Jain study, which seems to be the case. These data do not seem to provide an answer to the question of the accuracy of TR activity predictions, however.

- Knockdown: The HIF-1A knockdown experiment is an important addition to the work that does support their predictions, albeit in a single case.

In summary, the authors have added significantly more support for their findings. While there are still no comprehensive metrics of the accuracy of the TR activities predicted here, the work seems to be an important step and well-structured step in multi-omic integration for understanding regulation in cancer.

We sincerely appreciate the constructive comments and criticisms from the reviewer which we believe greatly helped us in maturing the manuscript.

Minor comments

The clarity around statistics was improved. I would still strongly suggest adding effect sizes where it is possible throughout the manuscript, for example to statements such as:

“For example, TRs involved in regulating cell cycle progression exhibit a significant tendency (p-value $\leq 1e-4$, Bonferroni-adjusted threshold) to correlate with the susceptibility to mTOR inhibitors, which in turn induce cell-cycle arrest.” The phrase “significant tendency to correlate” feels quite vague without knowing what value of correlation is significant, and neither the effect size nor statistical test was reported (or maybe the statistical test was in the methods and I failed to locate it).

We have carefully reviewed the manuscript and an adequate measure of effect size (i.e. mostly correlation analysis) is provided wherever needed.

My other remaining suggestion is that the authors offer their thoughts on how their work is affected by the completeness of the known TF-enzyme network, as was mentioned by another reviewer in the previous round.

We clarify this point in the discussion of the main text.

Reviewer #4:

None